# Sleep Architecture and EEG Power Spectrum Following Cumulative Sleep Restriction: A Comparison between Typically Developing Children and Children with ADHD

**DOI:** 10.3390/brainsci13050772

**Published:** 2023-05-08

**Authors:** Tamara Speth, Benjamin Rusak, Tara Perrot, Kimberly Cote, Penny Corkum

**Affiliations:** 1Department of Psychology & Neuroscience, Dalhousie University, Halifax, NS B3H 4R2, Canada; 2Department of Psychiatry, Dalhousie University, Halifax, NS B3H 4R2, Canada; 3Psychology Department, Brock University, St. Catharines, ON L2S 3A1, Canada

**Keywords:** cumulative sleep restriction, children, ADHD, sleep architecture, polysomnography

## Abstract

No studies have looked at the effects of cumulative sleep restriction (CSR) on sleep architecture or the power spectrum of sleep EEG (electroencephalogram) in school-age children, as recorded by PSG (polysomnography). This is true for both typically developing (TD) children and children with ADHD (attention deficit/hyperactivity disorder), who are known to have more sleep difficulties. Participants were children (ages 6–12 years), including 18 TD and 18 ADHD, who were age- and sex-matched. The CSR protocol included a two-week baseline and two randomized conditions: Typical (six nights of sleep based on baseline sleep schedules) and Restricted (one-hour reduction of baseline time in bed). This resulted in an average of 28 min per night difference in sleep. Based on ANOVAs (analysis of variance), children with ADHD took longer to reach N3 (non-rapid eye movement), had more WASO (wake after sleep onset) (within the first 5.1 h of the night), and had more REM (rapid eye movement) sleep than TD children regardless of condition. During CSR, ADHD participants had less REM and a trend toward longer durations of N1 and N2 compared to the TD group. No significant differences in the power spectrum were found between groups or conditions. In conclusion, this CSR protocol impacted some physiological aspects of sleep but may not be sufficient to cause changes in the power spectrum of sleep EEG. Although preliminary, group-by-condition interactions suggest that the homeostatic processes in children with ADHD may be impaired during CSR.

## 1. Introduction

In an increasingly fast-paced and technologically driven world, both children and adults are getting less sleep than ever before. Among children, there has been a decline in sleep duration of approximately one hour per night over the past century [1]. While the recommended sleep duration for children between 6 and 13 years of age is 9 to 11 h per night [2], a study of ~6000 children from 12 countries around the world found that only ~42% of children are meeting the expected sleep duration based on age guidelines [3]. The implications of this decline in sleep duration are especially troubling when one considers that sleep in childhood plays a critical role in neurodevelopment, learning, and the development of key functional skills [4]. Children with challenges in these key developmental skills, such as children with attention-deficit/hyperactivity disorder (ADHD), are at even greater risk for sleep problems, thereby exacerbating the impact of poor sleep on their daytime functioning [5].

There is a growing body of research about the negative impact of reduced sleep duration in children, both typically developing (TD) children and children with ADHD. Correlational studies have found a relationship between inadequate sleep (i.e., not meeting age-appropriate sleep recommendations), daytime sleepiness, difficulties with attention, impulse control, behavioural regulation, and impairments in cognitive functioning and academic performance [4]. Experimental sleep manipulation studies (in which sleep is experimentally manipulated by having the participant go to bed later), compared to correlational studies, are important as they allow for the inference of causation. Based on experimental sleep manipulation research, it has generally been found that sleep loss leads to impaired attention, increased restlessness and impulsivity, impaired academic performance, increased sleepiness (both subjective and objective), poorer emotion regulation, poorer neurocognitive functioning, and reduced alertness, relative to a baseline habitual or imposed 10 or 11 h time in bed (TIB) [6]. While there are fewer studies, research has found similar results for children with ADHD [6]. The impacts of sleep restriction in children are similar to those documented in the more extensive adult literature [7].

While research clearly demonstrates the impact cumulative sleep restriction (CSR) has on children’s daytime functioning, we do not know how CSR changes sleep physiology, including sleep architecture (i.e., structure of sleep including sleep stages and cycles) and the EEG power spectrum of sleep (i.e., micro-analysis of waveforms during sleep). This is important as sleep physiology is linked to the regulation of sleep-wake states and to different aspects of daytime functioning. Typically, studies looking at the effects of sleep loss on sleep physiology have taken two forms: acute sleep restriction or deprivation (much of the night or a full night of no sleep) and CSR (a milder form of sleep loss over multiple nights). In children, CSR represents a more real-world form of sleep loss [4], and as such, it is the focus of this research. Research on the impact of CSR on sleep physiology in adults is plentiful and points to a homeostatic response of sleep architecture (i.e., the pattern of sleep stages). This response typically takes the form of a preservation of N3 and a corresponding decrease in other stages of sleep, such as N2 and REM [7]. Studies in this area have also shown that changes in sleep are not adequately captured by sleep architecture alone, with authors noting changes in the power spectrum, including a dynamic increase in slow wave activity (SWA, activity in the delta band between 0.5 and 4 Hz) across all NREM stages. Some studies suggest that this increase may extend beyond the traditional delta band to include an increase in theta power [8]. It is thought that greater alpha or high-frequency beta in sleep represents abnormal arousal or brain activation in sleep, and greater theta or delta in NREM represents deeper sleep.

Similar findings have been found in adolescent samples [9,10,11] with the three existing studies finding reduced sleep onset latency (SOL) and wake after sleep onset (WASO), and two of the studies finding an increase in sleep efficiency (SE), suggesting an increase in sleep propensity due to the reduced sleep opportunity. All three studies also consistently noted a reduction in N1, N2, and REM sleep, with an increase in N3 during particular nights of the restriction protocol (there was no significant difference in the average duration of N3 across nights compared to baseline). Ong et al. [11] also found reduced REM latency across nights during restriction. Additionally, Ong et al. [11] looked at the first 5 h of sleep only (the longest common sleep duration across participants and conditions), as comparing a common temporal window may reveal changes in sleep homeostasis that would otherwise be undetected (this procedure has also been used in adult studies of CSR, e.g., Åkerstedt et al. [8]). They noted a reduction in the duration of N1 and an increase in REM and N3 relative to baseline. When Ong et al. looked at changes in SWA from baseline, they found that there was an increase in mean SWA across the full night. However, when SWA during the first 5 h of the night was compared between CSR and baseline nights, the mean SWA was maintained throughout the manipulation period. Therefore, Ong et al. [11] found that changes in sleep architecture were more apparent when comparing common sleep durations and that homeostatic changes in sleep were not fully reflected in sleep architecture alone.

There are currently no known studies that have investigated the effects of CSR in school-age (6–12-year-old) children on sleep architecture or the power spectrum of sleep EEG. Given that we know that children with ADHD are at greater risk for sleep problems and for the negative consequences of sleep problems on daytime functioning, it is important to consider whether the effects of sleep loss would be greater in children for whom sleep is already problematic. To address this research question, we accessed data collected during our previous study that examined the impact of CSR on daytime functioning in TD children (n = 18) and children with ADHD (n = 18) [12]. The past study utilized a seven-week repeated-measures protocol, which included a two-week baseline period, six nights of typical sleep (based on participants’ baseline sleep schedules), and six nights of restricted sleep (wherein TIB was reduced by one hour each night relative to participants’ average baseline sleep schedules). There was a one-week break following the baseline period, and Typical and Restricted conditions were counterbalanced and separated by a two-week recovery period. While TIB was reduced by approximately one hour, children compensated for this by falling asleep more quickly and having fewer night wakings, and as such, total sleep time only differed by approximately 20 min. Despite the short amount of sleep restriction, there were significant differences on an objective measure of attention and parent reports of emotional lability between the Typical and Restricted conditions across both the TD and ADHD groups [12].

Based on the findings presented above, the current exploratory study investigated changes in sleep architecture and the power spectrum of sleep EEG as a result of CSR in a sample of school-age children [12]. Looking at CSR (versus acute sleep restriction) is particularly relevant, as the former reflects a more typical pattern of sleep loss among children [4]. Furthermore, the current study compared physiological sleep processes during CSR between TD children and children with ADHD, a common neurodevelopmental disorder (NDD) in children that is known to be related to high levels of sleep problems [13]. Given that we know that children with ADHD are at greater risk for sleep problems and for the negative consequences of sleep problems on daytime functioning, it is important to consider whether the effects of sleep loss would be greater in children for whom sleep is already problematic.

## 2. Materials and Methods

### 2.1. Participants

TD children were recruited from the community using newsletters, web-based advertisements, and a research database of past research participants. Children in the ADHD group were recruited from a specialty ADHD clinic, two private practices focusing on NDDs, and through a research ADHD clinic, all of which employed the same diagnostic tools, including semi-structured diagnostic parent and teacher interviews, the collection of historical information, rating scales, and a psycho-educational assessment (for details, see McGonnell et al. [14]). Given that the DSM-V now specifies presentations of ADHD but no longer categorizes children with ADHD into subtypes, the current study included children across all ADHD presentations. All ADHD participants were medication-naïve for treatment of ADHD or any other psychopathology.

Before the study, potential participants were screened using questionnaires to ensure that they met all inclusion criteria and did not meet any of the general exclusion criteria. Inclusion criteria for both groups were that the children must be between 6 and 12 years of age. For the TD group, children could not have been previously diagnosed with a mental health disorder, whereas for the ADHD group, children were required to meet DSM-V diagnostic criteria for ADHD, to not have any comorbid mental health disorders (except for learning disorders), and to be medication naïve. These criteria were assessed through screening questionnaires for the TD group and a comprehensive clinical diagnostic assessment for the ADHD group. General exclusion criteria for the present study stipulated that participants must not have (1) a chronic and impairing medical illness, (2) a history of neurological impairments, (3) a primary sleep disorder (screened during the baseline PSG night in the laboratory), (4) used medication during the past month that is likely to affect sleep, (5) crossed more than two time zones in the last month, (6) regularly sleep less than 8 h or more than 12 h nightly, or (7) developed beyond Tanner stage 2 (based on a parent-completed questionnaire).

A total of 33 TD children and 32 children with ADHD between the ages of 6 and 12 met the initial screening criteria. Prior to the start of the baseline period, one TD child and five children with ADHD withdrew or were excluded by researchers. The one TD participant withdrew because they found the actigraph uncomfortable, and the five children with ADHD withdrew or were excluded for the following reasons: they could not make the schedule work, did not attend the baseline visit, met exclusion criteria, lost the actigraph, were sick, and did not give a reason. After enrollment in the baseline period, two TD children and four children with ADHD withdrew or were excluded from the study. The two TD participants were excluded because they met exclusion criteria; children with ADHD withdrew or were excluded for the following reasons: epileptic activity found on the PSG, diagnosis of ADHD could not be confirmed, could not make a time commitment, or did not give a reason. Therefore, 30 TD children and 23 children with ADHD completed the study protocol in its entirety. Of the children that completed all study weeks, seven TD children and five children with ADHD were excluded from the final analysis as they did not meet the sleep restriction criterion. This criterion required the average difference in TIB during the Restricted condition period and TIB during the Typical condition period as measured by actigraphy to be a minimum of 30 min less per night. One additional TD child was excluded from analyses due to the diagnosis of a chronic health disorder following the completion of the study. TD participants were then matched by age and sex with the ADHD participants, resulting in a final sample of 36 children (18 in each of the TD and ADHD groups). Consistent with the known higher incidence of ADHD among boys [15], the final sample included 14 boys and 4 girls in each group. A reduced sample was used for some analyses, as the EEG data for five participants (two boys with ADHD, two TD boys, and one TD girl) included a large amount of artifact that made some analyses impossible to complete, including power spectrum analysis. An additional age-matched girl with ADHD was removed from the analyses to ensure even numbers of participants in each group. Therefore, 15 participants (3 girls) remained in each group for these analyses. Sleep architecture analyses that were carried out using the first 5.1 h of sleep (the longest common sleep duration among all recordings, not including WASO) were also conducted with the reduced sample.

### 2.2. Procedure

During the baseline period, sleep durations were established for each participant over the course of two weeks, during which time they were asked to wear an actigraph (a wrist-worn accelerometer). Participants’ parents were also asked to complete a sleep diary during this time, in which they recorded various elements of the child’s sleep, such as bedtime, wake time, and number of awakenings during the night. At the end of this two-week period, participants spent a night in the sleep laboratory and had their sleep recorded using PSG (for details regarding PSG and resulting sleep variables, please see Jon 2009 [16]). After a one-week break, participants underwent a sleep manipulation protocol. Relative to baseline sleep patterns, participants were asked to go to bed one hour later nightly for one week (Restricted condition); during another week, their sleep schedule was assigned based on average baseline sleep and wake times (Typical condition). The order of the Restricted condition and Typical condition was counterbalanced, and there was a two-week recovery period between condition periods. Participants were also asked to wear an actigraph throughout each of these condition periods to confirm that they had followed the restriction protocol; adherence was confirmed following participation when actigraphy data were extracted and analyzed. Participants then came into the laboratory for an overnight PSG recording at the end of each of the condition periods, wherein sleep was scheduled according to their condition (i.e., they continued to follow their restriction schedule during the Restricted condition night in the laboratory and their typical sleep schedule during the Typical condition night in the laboratory).

#### 2.2.1. Polysomnography

PSG assessments were carried out by trained research assistants. Assessments were conducted with a Sandman^®^ PSG system, which recorded four electroencephalogram (EEG) channels (C3, C4, O1, O2), left and right electrooculograms (EOG), two submental electromyogram (EMG) channels, and an electrocardiogram (ECG). Leg movements were measured using electrodes placed on the left and right anterior tibialis muscles. Respiratory effort was measured using bands on the chest and abdomen, and breathing was measured using an oronasal cannula. Oxygen saturation was measured using a finger-probe pulse oximeter. Snoring was identified using a room microphone, and participants’ body positions were recorded using an infrared camera. Sleep stages were scored offline visually in 30-s epochs by a registered sleep technologist (supervised by a physician with a specialization in sleep medicine) according to American Academy of Sleep Medicine (AASM) guidelines [17]. The sleep technologist was blind to the condition of the PSG recordings. Sleep parameters (e.g., SOL, and SE) were calculated using reports generated through Sandman^®^. Normative values for comparison purposes can be found in Scholle et al. [18].

#### 2.2.2. Data Processing and Signal Analysis

PSG files were exported in European Data Format (EDF) and processed using Neuroscan version 4.5 SCAN software (Compumedics Neuroscan, Inc., El Paso, TX, USA). All files were either recorded at a sampling rate of 128 Hz or were down sampled to that value at the time of export. Using SCAN software, all EEG channels were re-referenced offline to the average of the two mastoid channels (A1/A2) for analysis, and signals were band-pass filtered between 1 and 30 Hz. Artifacts were then visually identified and highlighted. Using the reduced sample size (n = 15 per group), the average amount of artifact-free data from the Typical condition night was 89.22 percent and 91.47 percent from the Restricted condition night. The data were then exported into Microsoft Excel for the calculation of additional variables of interest.

EEG activity in human sleep ranges from about <1 Hz to 70 Hz and is typically described in bandwidths that have been shown to vary systematically with arousal and sleep depth. Absolute power values in the following bands were analyzed: delta (1–4 Hz); theta (4–7 Hz); alpha (8–12 Hz); sigma (13–16 Hz); and beta (16–24 Hz). Power spectrum analysis was conducted using fast Fourier transformation (FFT) with 2 s and 75% overlapped Hanning windows. Data were averaged between the left (C3) and right (C4) hemispheres and across NREM stages for the entire sleep period (i.e., all epochs of NREM that contained at least 15 s of continuous artifact-free EEG for the entire night) as well as for NREM within the first 5.1 h of sleep (i.e., epochs of NREM from the first 5.1 h of the EEG recording that contained at least 15 s of continuous artifact-free EEG). FFT analyses used a smaller sample size as they required a sufficient amount of artifact-free data. The data were logarithmically transformed prior to statistical analyses. In nine files, one channel had poor quality data for a portion of the sleep recording, and data from a single channel were used.

### 2.3. Data Analyses

To determine the impact of CSR on sleep parameters across the entire sleep period, two-way mixed ANOVAs were used to examine the effect of group (TD, ADHD) and experimental manipulation (Typical condition, Restricted condition). Separate ANOVAs were run for TIB, TST, SOL, WASO, and SE. To examine the effect of group and experimental manipulation on sleep architecture, two-way mixed ANOVAs were also used. Separate ANOVAs were run for minutes of N1, N2, N3, and REM. ANOVAs were also run to test for differences in sleep architecture using the first 5.1 h of sleep.

Looking at the power spectrum of sleep EEG, two-way mixed ANOVAs were used to examine the effect of group and experimental manipulation on SWA. Separate ANOVAs were run for NREM across the full night of sleep and NREM within the first 5.1 h of sleep. Exploratory analyses looking at the full power spectrum of sleep EEG were also conducted using two-way mixed ANOVAs. Separate ANOVAs were run for the full night of sleep and the first 5.1 h of sleep for each of the bands noted above.

Statistical assumptions were checked prior to running ANOVAs. Outliers were defined as studentized residuals plus or minus three values. For the sleep architecture data, variables that were non-normally distributed, contained significant outliers, and/or violated the assumption of homogeneity of variance were square root transformed. Transformations were applied to 14 out of 16 dependent variables. ANOVAs were run both prior to transformations being applied and after. As results did not change after applying transformations, results using the raw data are presented below. No further transformations were applied to FFT data, as these data were already logarithmically transformed prior to statistical analyses.

## 3. Results

### 3.1. Participants

No significant differences between groups were found in participants’ age, ethnicity, family composition, parental education, or family income as determined by *t*-tests and Pearson Chi-Squared tests. A *t*-test indicated a significant difference in estimated full-scale IQ between groups, with TD children having a higher IQ than children with ADHD, but both groups were in the average range (see Table 1). The study was run across several years throughout the school months (between September and June), with neither group being studied exclusively during any particular time of year.

### 3.2. Sleep Manipulation

There were main effects of condition, such that there was a statistically significant difference in TIB (F(1, 34) = 49.03, *p* < 0.001, partial η^2^ = 0.59) as well as in TST between conditions (F(1, 34) = 10.35, *p* < 0.01, partial η^2^ = 0.23), showing that, on average, participants spent 45.39 min less in bed and slept 28.37 min less during the Restricted condition night in the laboratory compared to the Typical condition night (see Table 2).

For all sleep parameters, there were no significant interaction effects. There were no main effects of condition or group on SE or SOL. There was no main effect of condition on WASO, but there was a main group effect (F(1, 34) = 4.54, *p* = 0.04, partial η^2^ = 0.118), such that children with ADHD had 15.21 min more WASO than TD children regardless of condition.

With regard to sleep architecture, there were no significant main effects for condition, group, or interactions for minutes of N1, N2, or N3. However, there was a trend towards an interaction for minutes of N1 (F(1, 34) = 3.26, *p* = 0.08, partial η^2^ = 0.09). Based on a visual inspection of the graphed and raw mean data for each group and condition, there appears to have been less N1 during CSR in the TD group and more N1 during CSR in the ADHD group. There was a significant group difference in latency to N3 sleep (F(1, 34) = 9.70, *p* = 0.004, partial η^2^ = 0.22), such that children with ADHD took 13.80 min longer to reach N3 than TD children regardless of condition. There was neither a significant main effect of the condition nor a significant interaction. There were no group differences in minutes of REM, but there was a statistically significant main effect of condition (F(1, 34) = 5.537, *p* = 0.025, partial η^2^ = 0.14), such that the amount of REM sleep was reduced by 14.94 min during restriction when compared to the Typical condition night in both groups. There was no significant interaction effect. There were no significant main effects for REM latency, but there was a trend towards a group-by-condition interaction (F(1, 34) = 3.917, *p* = 0.056, partial η^2^ = 0.103). Based on a visual inspection of the graphed and raw mean data for each group and condition, REM latency during CSR appeared longer in the ADHD group and shorter in the TD group.

Within the first 5.1 h of sleep (see Table 3), there were no main effects or interactions in minutes of N1. There were no group differences in the duration of N2, but there was a significant main effect of condition (F(1, 28) = 4.64, *p* = 0.04, partial η^2^ = 0.14), such that participants spent 11.55 more minutes in N2 during the Restricted condition night compared to the Typical condition night. There was also a trend towards a group-by-condition interaction (F(1, 28) = 3.21, *p* = 0.08, partial η^2^ = 0.10). Based on a visual inspection of the graphed and raw mean data for each group and condition, it seems that this condition effect was driven primarily by ADHD children, while there appears to have been very little change in N2 during CSR in TD children. There were no main effects or interactions for the duration of N3. Finally, there was a significant main effect of group for REM duration (F(1, 28) = 4.79, *p* = 0.04, partial η^2^ = 0.15), and a significant main effect of condition (F(1, 28) = 5.38, *p* = 0.03, partial η^2^ = 0.16), such that both groups had 10.90 fewer minutes of REM sleep during the Restricted condition night than during the Typical condition night, and that children with ADHD had 8.96 min more REM sleep than TD children, regardless of the experimental condition. There was no significant interaction effect.

There were no significant main effects or interactions in SWA or for power in any other bands investigated (see Table 4). In looking at the first 5.1 h of sleep, there were again no significant main effects or interactions in SWA or power in any other bands investigated. There was a trend towards a group-by-condition interaction for sigma (F(1, 28) = 3.44, *p* = 0.07, partial η^2^ = 0.11). Based on a visual inspection of the graphed and raw mean data for each group and condition, it appears that there was a greater amount of power in the sigma band during CSR in children with ADHD and less power in the sigma band during CSR in TD children.

## 4. Discussion

The current exploratory study is the first known study to experimentally investigate the effects of CSR in school-age children on both sleep architecture and the power spectrum of sleep EEG. Our study is also the first to investigate changes in these variables during CSR in children with ADHD, a clinical population known to have difficulty with sleep [13]. Results revealed a reduced duration of REM during CSR, with some differential changes in sleep architecture during restriction based on group. There were no significant findings regarding differences in the power spectrum of sleep EEG between groups or changes during CSR, with the exception of a trend towards a group-by-condition interaction for sigma power. Overall, our results suggest that there may be factors impacting homeostatic sleep processes in children with ADHD. These findings are preliminary, and more research needs to be done to understand the physiological response in children (both TD and ADHD) to varying degrees of CSR.

As expected, our study protocol led to a decrease in TST during the Restricted condition night across both groups, indicating that our experimental manipulation was successful. However, despite strict laboratory conditions, the mean difference in TST between Typical and Restricted condition nights was only 28 min. This is in contrast to TIB, which showed that children were in bed for an average of 45 min less during the Restricted condition night in the laboratory. Although there were no statistically significant changes in SOL, WASO, or SE, participants seem to be making up (to some degree) for the shortened TIB with more efficient use of their sleep opportunities. Looking at the raw data for these variables between conditions, the values for SE, SOL, and WASO are changing in the expected direction (i.e., higher SE, shorter SOL, and less WASO during CSR), with effect sizes for the condition effects in the medium range (0.04 to 0.07) [19]. It would therefore seem that CSR led to the expected changes in sleep parameters, although these changes were not large enough on the laboratory recording night to result in statistically significant differences between conditions. Our belief that participants were making up for the shortened sleep opportunity is supported by the finding that participants had a shorter SOL and decreased WASO (but no change in SE) during the Restricted condition period leading up to the laboratory night, as measured by actigraphy [6].

While our results did not show a maintenance of N3 with a corresponding reduction in all other stages of sleep, as was hypothesized, we did note a decrease in REM during CSR. The decrease in REM was also evident when we considered the first 5.1 h of sleep. This more pronounced deficit in REM is consistent with previous research [20] but is in contrast to our hypothesis based on the findings of Ong et al. [11], who noted an increased duration of REM when looking at the longest common sleep duration. The authors suggested that this may have been a result of their experimental protocol, which had participants go to bed two hours later during manipulation nights while aligning the midpoints of these manipulation nights with baseline sleep periods. Due to the circadian influence on REM sleep and therefore increased REM priority in the early morning/latter half of the sleep period, it is likely that the more dramatic delay in bedtime in Ong et al. resulted in a greater proportion of sleep falling within the morning circadian phase, contributing to the increase in REM that was not found in the current study [21]. Interestingly, when looking at the first 5.1 h of sleep only, we also noted an increase in N2. This finding is in contrast to the findings of Åkerstedt et al. [8] and Ong et al. [11], who found an increase of N3 during CSR and a decrease in other stages of sleep. Given that research in adults has shown a robust preservation in N3 and a reduction in N2 and REM following both acute and chronic sleep restriction (e.g., Banks & Dinges [7]), it is possible that the modest decrease in TST in the current study is what contributed, at least in part, to the maintenance of N2.

The lack of condition effect for N1 and the increase in N2 were both surprising but may be better understood within the context of the additional trends toward group-by-condition interactions for these variables. These trends suggest that, specifically, children with ADHD had more N1 and N2 during CSR. Therefore, for TD children, a decrease in N1 (full night), a maintenance in N2, and a decrease in REM (full night and first 5.1 h) are mostly consistent with our hypothesis that N3 sleep would be maintained at the expense of other stages of sleep. It is also worth noting that, while the ANOVA for the duration of N3 within the first 5.1 h of sleep did not come out as significant in the statistical analysis, the duration of N3 increased marginally (by approximately four minutes) early in the night in TD children, which is consistent with a homeostatic response to sleep restriction, while it decreased by approximately six minutes in children with ADHD. It is therefore unclear why children with ADHD would experience a decrease in N3 and an increase in N1 and N2 sleep during CSR. These findings are consistent with the group effect, which showed a longer onset of N3 in children with ADHD and suggest that there is something different about the homeostatic response to sleep restriction and the way in which N3 pressure accumulates in this group.

Given the increase in N2 and decrease in N3 during CSR in children with ADHD, it would seem that there is a prioritization of N2 sleep over N3 in this clinical population. This is consistent with our finding of a trend towards an increase in sigma power within the first 5.1 h of the night during CSR in children with ADHD. While N2 sleep is predominated by theta activity (4–7 Hz), sleep spindles are a distinguishing feature of this stage and are characterized by EEG activity within the sigma (13–16 Hz) range [22]. With the apparent increase in both N2 and sigma power during CSR in children with ADHD, it would be of interest to look at the impact of CSR on sleep spindles within this population.

These results should also be considered in light of our results for the full sleep period, which indicated that irrespective of condition, children with ADHD had more WASO. Although the apparent increase in N2 and decrease in N3 during CSR among children with ADHD (together with the finding that these children had a longer N3 latency and more WASO) may seem to indicate an altered homeostatic response, another possibility is that children with ADHD do experience the same homeostatic sleep processes as TD children but have a greater degree of difficulty reaching N3 sleep. This could be due to increased WASO interfering with their ability to reach this stage (thereby resulting in their increased latency to N3). Thus, the finding that children with ADHD spend more time in N2 sleep during CSR relative to TD children might indicate that while the brain is progressing towards achieving deeper sleep, arousals interrupt the progression from N2 to N3, thus increasing the time spent in N2 without these children reaching N3 as often. This interpretation is in line with the findings of previous researchers who examined sleep instability in children with ADHD by measuring the cyclic alternating pattern (CAP), a method of coding arousal fluctuations that do not lead to awakenings. The authors noted that children with ADHD had a reduced total A1 index [23,24] and a longer duration of A1 subtypes [24], which are responsible for the build-up and consolidation of slow wave NREM sleep and protecting sleep against disturbances [25]. Therefore, under typical sleep conditions, it is possible that children with ADHD are not experiencing the same degree of consolidated deep NREM sleep, which is further exacerbated and made more apparent under conditions of experimental sleep restriction.

This interpretation may also help to explain the trend towards an interaction for REM latency, such that REM latency appeared to be increased during CSR in the ADHD group and decreased during CSR in the TD group. Based on previous findings in youth [11,26] as well as findings in adult studies [27], we expected to see a decrease in REM latency during CSR. The decrease in REM latency is thought to reflect increased REM pressure accumulating over days of CSR as NREM sleep is prioritized [11,28]. While both groups were found to have a decrease in REM duration during CSR, given the finding that REM latency appeared to be increased during CSR in the ADHD group, it would seem that these children are not experiencing the same buildup of REM pressure. It may also be possible that repeatedly interrupted attempts to achieve N3 sleep during CSR resulted in this prolonged latency to REM, as proposed above.

Children with ADHD were also found to have more REM in the first 5.1 h of the night than TD children, regardless of their experimental condition. This finding leads one to consider the influence of the dopaminergic system as having an additional impact on sleep in these children. It has been proposed that both sleep problems and ADHD may stem from common etiologies such as altered dopaminergic pathways in the brain [13,29,30,31]. Dopamine has also been implicated as playing a role in the control of the sleep-wake cycle as well as the alternation between NREM and REM sleep, with dopaminergic neurons projecting to zones in the brain that are important for sleep-wake control [30,32], and there is research to suggest that lower levels of dopamine are related to increased REM propensity [33]. While there is some existing research to suggest that REM sleep is altered in children with ADHD, the direction of this relationship is inconsistent between studies and appears to depend on age and be modulated by sleep lab adjustment [30]. The relationship between ADHD, sleep disturbances, and the dopaminergic system is not yet fully understood and merits further research.

Our results did not reveal an increase in SWA during restriction, as would be expected based on previous research in both adolescents and adults [7,11]. Our lack of significant findings regarding SWA is in contrast to those of Ong et al. [11] but is consistent with at least one study with adults that failed to establish changes in SWA during CSR [20]. While the participants in Ong et al. [11] and Ong et al. [10] reported an average TIB of approximately 6 h on weeknights, they were required to adhere to a 9-h sleep schedule for one week prior to the experimental manipulation. The restriction to 5 h TIB for 5 or 7 nights thus results in a more extreme degree of restriction than was used in the current study, and therefore, similar to Skorucak et al. [20], it is possible that the manipulation used in the current study was too mild to lead to changes in sleep at the level of the power spectrum of the EEG. While more studies are needed to confirm this finding, it would seem to suggest that a reduction of TIB by about one hour less per night for six nights does not lead to a robust homeostatic response—both with regard to N3 and SWA—in school-age children. This finding highlights the fact that restricting TIB is not the same as restricting sleep, as individuals adapt to make up for some of this loss.

### 4.1. Clinical Implications

Our study suggests that children seem to adapt to a modestly shortened sleep opportunity, but it is unknown whether negative effects would accumulate if the restriction was continued. Given that the average TST during the Typical condition was more than one hour below the average amount recommended for this age group (8.67 h per night for TD children and 8.43 h for children with ADHD [2]), future studies may wish to examine the impact of the same degree of sleep restriction relative to a control period with a longer imposed sleep opportunity (rather than basing TIB during the Restricted condition off of participants’ typical sleep schedules). Additionally, as our results indicate that children with ADHD may not experience the same homoeostatic response to sleep loss as TD children or may have difficulty achieving deep sleep when sleep is restricted, the consequences of sleep loss for these children may be more severe. Future studies of experimental sleep restriction in this age group should consider the impact on children with ADHD relative to TD children, as was done by Gruber et al. [34].

### 4.2. Limitations

The current study had limitations that should be acknowledged. First, while the sleep manipulation employed reflected a realistic sleep deficit that may be experienced by school-age children, it is possible that one hour less TIB per night over a period of six nights is not sufficient to demonstrate robust changes in sleep architecture and the power spectrum of sleep. Furthermore, children in the current study were, on average, already sleeping approximately one hour less than the average recommended amount for this age group. It is therefore possible that our participants had already adapted to a reduced sleep duration. It should also be highlighted that the current study had a modest sample size, which was reduced further for some analyses. This likely impacted our ability to detect effects and limited our confidence in our results. Therefore, the findings presented above should be thought of as hypothesis-generating as they provide preliminary evidence of altered homeostatic processes in children with ADHD and offer possible directions for future lines of inquiry. Finally, the current study only considered the impact of CSR on medication naïve children with ADHD without any comorbid mental health disorders; this is not representative of many children with ADHD. However, given that this study is the first to investigate the effects of CSR on sleep architecture and the power spectrum of sleep EEG in this population, it was important to establish the impact of CSR using an unmedicated sample with no comorbidities. Future studies may wish to explore the generalizability of these results.

### 4.3. Areas for Future Research

Given that the current study is the first to investigate the effects of CSR on sleep architecture and the power spectrum of sleep EEG in TD school-age children and children with ADHD, future studies should determine dose-response effects of CSR within this population (similar to Van Dongen et al. [27]). Given the limited findings in our study, it is possible that one hour less TIB per night over six nights is not sufficient for changes to be reflected in sleep physiology (particularly the power spectrum of the EEG). It will be important to determine the dose at which a homeostatic response is detectable and if/how this response changes with varying degrees of restriction. It may also be of interest to look at differences in sleep spindles given that sleep spindles are impacted by sleep loss [35], and there is some research pointing to differences in sleep spindles/sigma power in sleep EEG between TD children and children with ADHD [36,37]. It would be particularly interesting to look at changes in sleep spindles in these groups during CSR given the trends suggesting a greater amount of N2 and sigma power in children with ADHD during CSR in the current study. Finally, it will be important to further study the impact of both sleep restriction and extension, especially given that children are getting less sleep at baseline than is recommended.

## 5. Conclusions

The first goal of the current study was to (1) investigate how school-age children respond to mild CSR with regard to physiological changes in sleep. We noted less TIB and TST during the Restricted condition, with no change between conditions in SOL, SE, or WASO. Regarding sleep stages, there was a main effect of a shorter duration of REM during CSR. During the first 5.1 h of sleep, we again noted a shorter duration of REM as well as a longer duration of N2 during CSR relative to sleep during the Typical condition. There were no differences in SWA in either analysis. The second and third goals of the current study were (2) to investigate whether there were differences in sleep between TD children and children with ADHD regardless of condition, and (3) to determine whether children with ADHD would be more severely impacted by CSR compared to TD children. Our findings indicated that across conditions, children with ADHD had more WASO and a longer latency to N3, as well as more REM within the first 5.1 h of the night. There were no differences between groups in SWA or any of the additional frequency bands examined. Trends towards interaction effects suggested that children with ADHD had more minutes of N1 and a longer REM latency during the Restricted condition compared to the Typical condition, while values for TD children appeared to change in the opposite direction (i.e., fewer minutes of N1 and a shorter latency to REM during CSR). Within the first 5.1 h of the night, we found a trend suggesting that the increase in N2 during CSR was primarily driven by children with ADHD. Similar to TD children, there were no differences in mean SWA during CSR, either when looking at the full night or the first 5.1 h of sleep. The final goal of the current study was to explore changes in the power spectrum of sleep beyond the traditional delta band. We found no differences between groups in power within the theta, alpha, or beta bands, either between groups or between conditions. There was a trend towards greater sigma power during CSR in the first 5.1 h of sleep in the ADHD group, with what appeared to be less sigma power in the TD group.

## Figures and Tables

**Table 1 brainsci-13-00772-t001:** Participant Demographics by Group.

	ADHD (*n* = 18)	TD (*n* = 18)		
Variable	M (SD)	M (SD)	F	*p*
Age (months)	107.67 (18.96)	104.06 (16.43)	0.37	0.55
Estimated FSIQ	99.92 (10.86)	107.98 (9.50)	5.62	0.02
Variable			χ^2^	*p*
Ethnicity (*n* = 33)	15/16 White/Caucasian;	15/16 White/Caucasian;	0.00	1.00
	1/16 Multi-Racial	1/16 Multi-Racial		
Family Composition (*n* = 33)	13/17 two parent;	16/17 two parent;	2.11	0.15
	4/17 single parent	1/17 single parent		
Maternal Education (*n* = 28)	6.00	4.50	4.70	0.45
Paternal Education (*n* = 30)	2.50	4.00	6.48	0.26
Annual Family Income (*n* = 36)	5.00	6.00	4.86	0.43

Note: Age in years and months at the time of baseline; M = mean; SD = standard deviation; TD = typically developing; FSIQ = full-scale intelligence quotient. Median values presented for maternal and paternal education: 1 = some secondary/high school; 2 = completed secondary/high school; 3 = some community, technical, or CEGEP college; 4 = completed community, technical, or CEGEP college; 5 = some university or teacher’s college; 6 = completed university or teacher’s college. Median values presented for annual family income: 1 = up to $20,000; 2 = $20,001 to $30,000; 3 = $30,001 to $40,000; 4 = $40,001 to $50,000; 5 = $50,001 to $60,000; 6 = $60,001 to $70,000; 7 = more than $70,000.

**Table 2 brainsci-13-00772-t002:** Sleep Parameters and Sleep Architecture Across the Full Sleep Period.

	Mean (SD)						
	ADHD (*n* = 18)		TD (*n* = 18)		Effect Size *		
	Typical	Restricted	Typical	Restricted	G	C	G × C
TIB (min)	575.15 (37.64)	535.06 (37.75)	590.55 (43.91)	539.87 (39.33)	0.02	0.59 *	0.02
TST (min)	505.67 (56.09)	473.51 (53.85)	519.91 (57.01)	495.34 (39.78)	0.04	0.23 *	0.01
SE (%)	87.87 (7.08)	88.41 (6.87)	88.19 (8.86)	91.75 (3.15)	0.03	0.06	0.03
SOL (min)	19.94 (15.17)	17.27 (19.63)	25.56 (30.39)	15.89 (11.01)	0.004	0.07	0.02
WASO (min)	49.14 (41.28)	43.92 (32.06)	37.36 (26.60)	25.28 (14.76)	0.12 *	0.04	0.01
N1 (min)	26.74 (10.27)	31.11 (16.87)	30.24 (11.54)	25.43 (10.39)	0.003	<0.001	0.09~
N2 (min)	223.77 (57.26)	219.26 (40.92)	231.06 (42.75)	227.97 (36.33)	0.01	0.01	<0.001
N3 (min)	140.68 (45.44)	123.97 (39.80)	144.50 (39.94)	142.39 (23.37)	0.03	0.05	0.03
N3 Latency (min)	13.67 (11.51)	15.14 (13.01)	8.36 (2.99)	9.56 (3.95)	0.22 *	0.01	<0.001
R (min)	114.49 (20.73)	99.18 (30.28)	114.11 (30.02)	99.54 (24.64)	<0.001	0.14 *	<0.001
R Latency (min)	119.39 (42.12)	138.61 (43.55)	130.17 (53.32)	113.53 (44.93)	0.01	0.001	0.10~

Note: SD = standard deviation; ADHD = attention-deficit/hyperactivity disorder; TD = typically developing; TIB = time in bed; TST = total sleep time; SE = sleep efficiency; SOL = sleep onset latency; WASO = wake after sleep onset; N1–N3 = non-REM sleep stages 1–3; R = REM sleep; G = group main effect; C = condition main effect; G × C = group by condition interaction. * = significant at *p* < 0.05; ~ trend at *p* > 0.05 and < 0.10.

**Table 3 brainsci-13-00772-t003:** Sleep Architecture Within the First 5.1 h of Sleep (Table 2).

	Mean (SD)						
	ADHD (*n* = 15)		TD (*n* = 15)		Effect Size *		
	Typical	Restricted	Typical	Restricted	G	C	G × C
N1 (min)	8.44 (4.03)	10.20 (4.89)	10.67 (4.51)	9.50 (6.05)	0.01	0.003	0.06
N2 (min)	119.06 (39.90)	140.23 (39.09)	131.06 (19.18)	133.00 (19.37)	0.003	0.14 *	0.10~
N3 (min)	118.03 (34.84)	112.37 (31.25)	119.14 (23.04)	122.90 (18.65)	0.02	0.001	0.03
R (min)	62.47 (24.80)	45.20 (8.72)	47.14 (15.20)	42.60 (15.61)	0.15*	0.16 *	0.06

Note: SD = standard deviation; ADHD = attention-deficit/hyperactivity disorder; TD = typically developing; N1–N3 = non-REM sleep stages 1–3; R = REM sleep; G = group main effect; C = condition main effect; G × C = group by condition interaction. * = significant at *p* < 0.05; ~ trend at *p* > 0.05 and <0.10.

**Table 4 brainsci-13-00772-t004:** The Power Spectrum of NREM.

	Mean (SD)						
	ADHD (*n* = 15)		TD (*n* = 15)		Effect Size *		
Full Night (NREM)	Typical	Restricted	Typical	Restricted	G	C	G × C
SWA	3.37 (0.25)	3.42 (0.25)	3.40 (0.24)	3.42 (0.25)	0.001	0.02	0.01
Theta	2.31 (0.18)	2.37 (0.37)	2.38 (0.28)	2.36 (0.20)	0.01	0.004	0.02
Alpha	1.60 (0.20)	1.71 (0.52)	1.73 (0.41)	1.67 (0.27)	0.01	0.004	0.04
Sigma	1.00 (0.27)	1.18 (0.56)	1.19 (0.49)	1.06 (0.29)	0.004	0.003	0.09
Beta	0.76 (0.20)	0.94 (0.61)	0.98 (0.60)	0.81 (0.26)	0.01	<0.001	0.08
First 5.1 h (NREM)							
SWA	3.43 (0.27)	3.49 (0.24)	3.49 (0.25)	3.49 (0.27)	0.004	0.01	0.01
Theta	2.35 (0.19)	2.41 (0.38)	2.44 (0.27)	2.40 (0.21)	0.01	<0.001	0.04
Alpha	1.63 (0.21)	1.74 (0.53)	1.77 (0.41)	1.70 (0.27)	0.01	0.002	0.05
Sigma	0.97 (0.25)	1.17 (0.58)	1.17 (0.50)	1.02 (0.28)	0.001	0.002	0.11~
Beta	0.75 (0.21)	0.92 (0.63)	0.96 (0.60)	0.78 (0.25)	0.003	<0.001	0.08

Note: SD = standard deviation; ADHD = attention-deficit/hyperactivity disorder; TD = typically developing; SWA = slow wave sleep; Hrs = hours; G = group main effect; C = condition main effect; G × C = group by condition interaction. * = significant at *p* < 0.05; ~ trend.

## Data Availability

Data are not available due to ethical restrictions.

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
