# Peer review of "Sleep Architecture and EEG Power Spectrum Following Cumulative Sleep Restriction: A Comparison between Typically Developing Children and Children with ADHD"

_brainsci, 2023, doi:10.3390/brainsci13050772_

Round 1

Reviewer 1 Report

ID: brainsci-2336911

Title: Sleep Architecture and EEG Power Spectrum following Cumulative Sleep Restriction: A Comparison between Typically Developing Children and Children with ADHD

Thank you for providing a chance to review this manuscript.

Detailed information:

Abstract

Line 15~16, Page 1: One point in the Abstract confuses me: why does it suddenly appear that the subjects of the study are children with TD and children with ADHD? I would suggest you clearly list the aim of your study in one sentence to make the Abstract clearer and more logical.

Line 18~20, Page 1: The Abstract does not clearly demonstrate the analytical methods used in your study, and I suggest you change this section.

Line 24~26, Page 1: “In conclusion, this CSR protocol impacted some physiological aspects of sleep but may not be sufficient to cause changes in the power spectrum of sleep EEG”, which of the supporting data you have shown leads to this conclusion?

Overall:

(1) My small suggestions to you are to chunk the Abstract and add subheadings such as background, methods, results, and conclusions, and check for complete and concise descriptions after each subheading.

(2) Should there be an explanation of abbreviations such as EEG, PSG, ADHD, N3, WASO, and so on, which are not common terms and which appear for the first time in the text?

(3) The results should show the main statistics of the study.

Introduction

Line 39~41, Page 1: Is chronic inadequate a disease condition that is subordinate to sleep deprivation? Now that you mention it, it would be best to give a straightforward definition.

Line 31~55, Page 1~2: (1) Why do you repeatedly refer to adult sleep several times in the main text when your study population is children? If you want to make a cross-sectional comparative reference, it would be advisable to give the necessary explanation at the beginning or ending separately, rather than passing over it casually; (2) “However, additional research is needed to corroborate and expand on these findings”, this is a very vague statement, which aspect of each needs to be corroborated and expanded?

Line 53~55, Page 2: “These studies are important as they allow for inference of causation; however, additional research is needed to corroborate and expand on these findings”. What is the inference of causation? Please expand on it.

Line 56~57, Page 2: Sleep restriction is an extremely important definition relevant to your study and I suggest you write it clearly at the beginning.

Overall:

(1) You have given a very lengthy and extensive literature support in the Introduction section, I suggest you cut it down by removing descriptions that are not relevant to your research topic and some excessive literature examples, appropriate and closely related literature support is what is desirable.

(2) The clinical implications and effects of sleep architecture and EEG power spectrum need to be introduced, and the reasons for choosing these two indicators in this study should be explained.

(3) Why were children with ADHD selected as study subjects? What is the status of this population?

(4) Based on your literature review, I suggest you give your research innovation and hypothesis.

Materials and Methods

Participants

Line 170~187, Page 4: I would suggest you add a flowchart to show the sample size related data more visually.

Line 170~176, Page 4: The loss of follow-up rate in this study was too high, and I suspect the recruitment procedure was flawed.

Line 188~189, Page 4: (1) How was the matching of age and gender between the two groups done? I would like to see a more detailed description; (2) Your final sample composition was 14 males and 4 females, is this consistent with the gender profile of the study group? This could lead to gender bias; (3) Only 36 participants were included in this study, the sample size is too small. Healthy controls there are so few, and it's incredible. Are there sample size calculations to support that your included study population is adequate?

Line 189~195, Page 4: This section is more suitable for "Results".

Procedure

Line 208~236, Page 4~5: This part is too redundant and can be described in layers according to the experimental time and method.

Data Analyses

Normal levels of sleep architecture and EEG power spectrum are expected to be listed.

Overall:

(1) Given that this study is an intervention with subjects, I suggest you add a description of informed consent, compliance with ethical principles, etc.

(2) The method description process is too complicated, and the author should use more understandable language to describe the research process. In addition, the language needs to be reorganized to highlight the focus of research methods.

(3) The author spent a lot of time describing the research process, but there is little explanation of the observed indicators, please add relevant content.

Results

Line 304~306, Page 7: “participants spent 45.39 minutes less in bed and slept 28.37 minutes less during the Restricted condition night in the laboratory compared to the Typical condition night”. How is this difference calculated? Which population does the data represent? These results are not presented in table 2.

Line 308, Page 7: “There was no main effect of group and no condition by group interaction.” What does this phrase mean? How to come to this conclusion?

Line 342~358, Page 8: For the description of this paragraph, it is still recommended to indicate the specific statistical values results.

Line 352, Page 8: The results of “The Power Spectrum of NREM” are not found in Table 2.

Line 361, Page 8: Please present the results of follow-up in table form.

Table 2: There are missing abbreviations in this table, please check and add them.

Overall: The results were described in a confusing manner, and the author is supposed to rearrange the language to make the expression more logical.

Discussion

Line 415~416, Page 10: “We found no differences in power within the theta, alpha, or beta bands either between groups or between conditions.” How this conclusion was arrived at? And it is recommended that the authors present the key results in the form of graphs.

Overall: I have to say that the discussion is too lengthy that readers may have no desire to read carefully. It is recommended that the author sort out the logic, streamline the conclusions of the research, and discuss the core content.

To be honest, the research process is relatively simple, but the author spends a lot of time on the language, which makes it difficult for the reader to grasp the main point of the research. This study is committed to analyze and compare separately sleep architecture and EEG power spectrum of typically developing children and children with ADHD following cumulative sleep restriction, which has strong application values in the behavioral traits yield. First of all, the sample size of this study is insufficient and the loss of follow-up rate is high, so I suspect that there is something wrong with the recruitment procedure. Such a small sample size is not enough to convince me of the conclusion of this study, so I suggest the author to further expand the sample size. In addition, it is very difficult to read the text description of the whole article, though I admit that the authors described it very carefully. I cannot extract the key content, and it is suggested the author read high-quality articles and rearrange the language. Last but not least, there are many small mistakes in this article, such as abbreviations and so on. Look forward to seeing this research more perfect presentation!

Thank you and my best,

Your reviewer

Author Response

ABSTRACT

Line 15~16, Page 1: One point in the Abstract confuses me: why does it suddenly appear that the subjects of the study are children with TD and children with ADHD? I would suggest you clearly list the aim of your study in one sentence to make the Abstract clearer and more logical.

We understand the concern and have added some text to include ADHD in the rationale. It was not possible to add more information as the Abstract has a 200 word limit, and it is currently at max.

Line 18~20, Page 1: The Abstract does not clearly demonstrate the analytical methods used in your study, and I suggest you change this section.

We added that ANOVAs were used. We could not include more details due to word limits.

Line 24~26, Page 1: “In conclusion, this CSR protocol impacted some physiological aspects of sleep but may not be sufficient to cause changes in the power spectrum of sleep EEG”, which of the supporting data you have shown leads to this conclusion?

Again, with the word limit being 200, it is not possible to include details to support conclusions. These are found in the Discussion.

Overall:

  • My small suggestions to you are to chunk the Abstract and add subheadings such as background, methods, results, and conclusions, and check for complete and concise descriptions after each subheading.

In the instructions to authors it was noted to not use headings in the abstract. However, the abstract now includes information in all key sections including background, methods, results, and conclusions.

  • Should there be an explanation of abbreviations such as EEG,PSG, ADHD, N3, WASO, and so on, which are not common terms and which appear for the first time in the text?

This is not possible in the abstract but we ensured that this was done in the main body of the paper.

  • The results should show the main statistics of the study.

The main findings are covered in the Abstract.

INTRODUCTION

Line 39~41, Page 1: Is chronic inadequate a disease condition that is subordinate to sleep deprivation? Now that you mention it, it would be best to give a straightforward definition.

This sentence was removed as it was repetitive with the sentence just prior and it was suggested by the current reviewer to streamline the Introduction. We tried to streamline while also maintaining the core content as another reviewer noted that they appreciated the extensive literature review.

Line 31~55, Page 1~2: (1) Why do you repeatedly refer to adult sleep several times in the main text when your study population is children? If you want to make a cross-sectional comparative reference, it would be advisable to give the necessary explanation at the beginning or ending separately, rather than passing over it casually; (2) “However, additional research is needed to corroborate and expand on these findings”, this is a very vague statement, which aspect of each needs to be corroborated and expanded?

We have tightened the section about the impact of sleep loss on children and made it clear why experimental sleep manipulation studies were important (i.e., provide causal information unlike correlational studies). This section was substantially revised so that it is clear why it is important to review the findings of experimental sleep restriction studies on sleep physiology in adults and adolescents. Given that there is no research of this kind in children, the comparison is critical to help guide interpretation of the results in child-focused research.

Line 53~55, Page 2: “These studies are important as they allow for inference of causation; however, additional research is needed to corroborate and expand on these findings”. What is the inference of causation? Please expand on it.

This sentence was removed as it wasn’t relevant for building the rationale for this study.

Line 56~57, Page 2: Sleep restriction is an extremely important definition relevant to your study and I suggest you write it clearly at the beginning.

This has been defined.

Overall:

  • You have given a very lengthy and extensive literature support in the Introduction section, I suggest you cut it down by removing descriptions that are not relevant to your research topic and some excessive literature examples, appropriate and closely related literature support is what is desirable.

We tried to streamline while also maintaining the core content as another reviewer noted that they appreciated the extensive literature review.

  • The clinical implications and effects of sleep architecture and EEG power spectrum need to be introduced, and the reasons for choosing these two indicators in this study should be explained.

We restructured the Introduction to highlight the rationale for this study and for the focus on sleep architecture and power spectrum.

  • Why were children with ADHD selected as study subjects? What is the status of this population?

We restructured the Introduction so that the rationale for including children with ADHD was made earlier and more clearly, and now references the fact that children with ADHD are at increased risk of sleep problems and resulting daytime sequel.

(4) Based on your literature review, I suggest you give your research innovation and hypothesis.

Hopefully the innovative nature of this research is now clearer with the restructuring of the Introduction. We have also added a general hypothesis regarding the differential impact of chronic sleep restriction on children with ADHD.

MATERIALS AND METHODS

Participants

Line 170~187, Page 4: I would suggest you add a flowchart to show the sample size related data more visually.

A flowchart was prepared and is now in the Supplemental Materials.

Line 170~176, Page 4: The loss of follow-up rate in this study was too high, and I suspect the recruitment procedure was flawed.

Most participants were excluded due to pre-established criteria, with less than 30% of participants choosing not to continue in this study. The drop-out rate is similar to other experimental sleep manipulation studies given that these are highly demanding studies on the child and family’s time (e.g., required 3 overnights at a sleep lab with full polysomnography, plus a full day of assessment battery; parents stayed in an adjacent room). The fact that most of the participants who choose not to participate were in the ADHD group is not surprising, given that these families often are experiencing higher levels of stress making this demanding research protocol more challenging for these families.

Line 188~189, Page 4: 

  • How was the matching of age and gender between the two groups done? I would like to see a more detailed description;

A brief description has been added.

  • Your final sample composition was 14 males and 4 females, is this consistent with the gender profile of the study group? This could lead to gender bias;

Sex ratios for ADHD children are between thought to be around 1:3 (1 female for every 3 males) in children. As such our ratio of 4:14 (4 females and 14 males or ~1:3) is consistent with the sex ratio statistics in the literature. Moreover, for prepubertal children, there are minimal sleep differences found (reference).

  • Only 36 participants were included in this study, the sample size is too small. Healthy controls there are so few, and it's incredible. Are there sample size calculations to support that your included study population is adequate?

The sample size is consistent with other studies with such rigorous and time intensive research protocols. Moreover, given that it was an exploratory study, there was no way to estimate the potential effect sizes needed to calculate power. We appreciate and acknowledge this limitation in the Discussion section.

Line 189~195, Page 4: This section is more suitable for "Results".

    Information about the participants and Table 1 were moved to the Results section.

Procedure

Line 208~236, Page 4~5: This part is too redundant and can be described in layers according to the experimental time and method.

After reflecting on this feedback, we decided to leave the information about procedures in this section rather than integrating them elsewhere as we think this is easier for the reader.

Data Analyses

Normal levels of sleep architecture and EEG power spectrum are expected to be listed.

A reference has been added that provides normative values for sleep architecture parameters for children. There are no normative EEG power values used diagnostically in sleep. The data processing and signal analysis section for the EEG power spectrum was revised to be clearer.

Overall:

  • Given that this study is an intervention with subjects, I suggest you add a description of informed consent, compliance with ethical principles, etc.

This information is included at the end of the manuscript under “Institutional Review Board Statement” and as such we thought it did not need to be repeated in the text.

  • The method description process is too complicated, and the author should use more understandable language to describe the research process. In addition, the language needs to be reorganized to highlight the focus of research methods.

The Methods section has been revised in response to the reviewer’s comments. As well, all acronyms have been defined. We hope that we have made this section more reader-friendly.

  • The author spent a lot of time describing the research process, but there is little explanation of the observed indicators, please add relevant content.

We appreciate that the methods section is technical given the nature of the research. We have tried to make it clearer when possible and defined all acronyms. We have also included a reference to a book chapter that provides details about polysomnography and the resulting sleep parameter variables.

RESULTS

Line 304~306, Page 7: “participants spent 45.39 minutes less in bed and slept 28.37 minutes less during the Restricted condition night in the laboratory compared to the Typical condition night”. How is this difference calculated? Which population does the data represent? These results are not presented in table 2.

These results were included to demonstrate that the sleep manipulation was successful and resulted in ~30 min less sleep between the typical and restricted sleep conditions. This was calculated based on the difference between typical and restricted condition nights for all participants.

Line 308, Page 7: “There was no main effect of group and no condition by group interaction.” What does this phrase mean? How to come to this conclusion?

This was removed as it was related to actigraphy data and as such not relevant to this study.

Line 342~358, Page 8: For the description of this paragraph, it is still recommended to indicate the specific statistical values results.

This was removed as it was related to actigraphy data and as such not relevant to this study.

Line 352, Page 8: The results of “The Power Spectrum of NREM” are not found in Table 2.

Thank you for finding this omission. We have included this data in Table 3.

Line 361, Page 8: Please present the results of follow-up in table form.

The follow-up analyses have been removed.

Table 2: There are missing abbreviations in this table, please check and add them.

We added definitions for all abbreviations.

Overall: The results were described in a confusing manner, and the author is supposed to rearrange the language to make the expression more logical.

We decided the follow-up analyses were not relevant to the overall focus of this manuscript and as such deleted these. This should simplify the presentation of the results and reduce any potential confusion.

DISCUSSION

Line 415~416, Page 10: “We found no differences in power within the theta, alpha, or beta bands either between groups or between conditions.” How this conclusion was arrived at? And it is recommended that the authors present the key results in the form of graphs.

To address feedback from another reviewer, we exchanged the first paragraph of the Discussion with the Conclusion section. We also clarified that the sentence above refers to differences between groups. We also highlighted that this was an exploratory study in which there were some interesting findings that need to be followed up on with additional research. We have also added in the interpretation of differences in EEG power spectrum in the Introduction, which should help with this information in the Discussion. In terms of graphs, we do not think graphing insignificant results would add any additional information.

Overall: I have to say that the discussion is too lengthy that readers may have no desire to read carefully. It is recommended that the author sort out the logic, streamline the conclusions of the research, and discuss the core content.

To be honest, the research process is relatively simple, but the author spends a lot of time on the language, which makes it difficult for the reader to grasp the main point of the research. This study is committed to analyze and compare separately sleep architecture and EEG power spectrum of typically developing children and children with ADHD following cumulative sleep restriction, which has strong application values in the behavioral traits yield. First of all, the sample size of this study is insufficient and the loss of follow-up rate is high, so I suspect that there is something wrong with the recruitment procedure. Such a small sample size is not enough to convince me of the conclusion of this study, so I suggest the author to further expand the sample size. In addition, it is very difficult to read the text description of the whole article, though I admit that the authors described it very carefully. I cannot extract the key content, and it is suggested the author read high-quality articles and rearrange the language. Last but not least, there are many small mistakes in this article, such as abbreviations and so on. Look forward to seeing this research more perfect presentation!

Thank you for your thorough review of this manuscript. We believe that addressing your feedback has made this a strong and clearer written manuscript. We have streamlined the Introduction and Discussion, and ensured that terms were defined. With the utmost respect, we do not agree that this protocol is simple, and as such, left the detailed procedures, which were noted as appropriate by another reviewer. Experimental sleep manipulation studies are extremely labour intensive and as such it is typical to have small sample sizes. Also, exclusion criteria were rigorous which allowed us to have strong internal validity, albeit, potentially reducing external validity. We have also clearly indicated that this is an exploratory study, which addresses a highly novel and needed area of study. This study found intriguing results and information which can be used to further explore the impact of sleep restriction on sleep physiology in children (e.g., information to allow for power analyses for sample size calculations). We hope that we have been able to address your main concerns and that you now find the manuscript to be more streamlined and clearer, and that the innovative aspect of this research is now more salient.

Reviewer 2 Report

"Sleep Architecture and EEG Power Spectrum following Cumulative Sleep Restriction: A Comparison between Typically Developing Children and Children with ADHD" is a very interesting and well prepared study.

The summary and the references could be improved.

Author Response

Thank you for your positive review. We reviewed the entire manuscript and made any necessary changes including streamlining where possible (resulting in fewer references) and revising the conclusion.

Reviewer 3 Report

The manuscript explores data from EEG studies on children (6-12) subjected to typical sleep as well as cumulative sleep restriction (CSR). Children were split between typically developing children (TD) and children with ADHD. For both typical and CSR conditions, children with ADHD had more WASO and a longer latency to N3, as well as more REM within the first 5.1 hours. 

Restricted conditions lowered TIB, and TST in both children groups. Also, the duration of N1 and REM latency was increased in children with ADHD, as opposed to typically developing children, for whom these quantities followed the opposite trend. Also, an increase in N2 was observed among children with ADHD. There is also an indication of a greater sigma power during CSR in the ADHD group as opposed to the TD group.

The article puts forth an important question regarding the difference in sleep architecture and the effect of sleep deprivation in children with ADHD. This study can be considered a preliminary attempt at getting into the details of such an understanding with ample scope to expand on the cues seen here. Indeed, as pointed out the sample size is modest, and the work does not explore situations of extreme sleep restrictions, it already hints at different adaptability in the case of children with ADHD.

The article begins with an excellent review of the knowledge base on the subject, however, the discussion section could use some brevity. The first paragraph of the discussion could be treated as the conclusion of the article and can be merged with the current conclusion.

Author Response

Thank you for your positive review. We are pleased that you find the results to be intriguing, pointing to some areas of future exploration. We have switched the conclusion paragraph to now start the Discussion and the previous first paragraph is now the conclusion. We also reviewed and edited the Discussion to make it more succinct where possible. We hope that this revised manuscript meets with your approval.

Round 2

Reviewer 1 Report

ID: brainsci-2336911

Title: Sleep Architecture and EEG Power Spectrum following Cumulative Sleep Restriction: A Comparison between Typically Developing Children and Children with ADHD

Detailed information:

Abstract

Overall: The abstract is an important part of the article, which directly determines whether the readers will read the full text. The length limit of abstracts is a requirement for almost all magazines, and this is not a reason for vague description. The Abstract needs to describe your theoretical foundations, research design, and presentation of results in the shortest content. You should reorganize the description of the Abstract to bring it closer to the more critical content, rather than using word limits as a reason to refuse to add descriptions.

Introduction

Overall: 1) The author directly deleted the sentence with doubt, which is not my purpose, but needs a more specific and convincing explanation; 2) I have no problem with the other reviewer appreciated the extensive literature review. Nevertheless, extensive literature reviews and lengthy extraneous descriptions are not in conflict, and I still expect a more concise and focused description.

Methods

Overall: (1) For explanations in the Author Response, please also describe them clearly in the manuscript; (2) Since the high loss of follow-up rate is a common problem of similar studies, the author should take it into account when calculating the sample size at the initial stage of the study and recruit a larger sample.

Results

Overall: What makes the reader’s reading difficult is the confusing logic of expression and the lack of concise linguistic descriptions; it is not reasonable to fundamentally remove the research design, and follow-up still has its research relevance.

The author made some changes to the article, but it still did not address the main doubts. First of all, the author should give comprehensive consideration to the opinions of each reviewer. It is normal that different reviewers have different opinions. The author should not use another reviewer's opinion to reject my opinion, but should list more authoritative evidence to convince me. In addition, although this article is an exploratory study, the author did not explain in detail the problem of small sample size and high loss to follow-up, which did not convince me. Finally, the author is supposed to highlight the modified part to better represent the traces of modifications. 

Thank you and my best,

Your reviewer

Author Response

Reviewer 1 Comments on Revised Manuscript

Abstract

Overall: The abstract is an important part of the article, which directly determines whether the readers will read the full text. The length limit of abstracts is a requirement for almost all magazines, and this is not a reason for vague description. The Abstract needs to describe your theoretical foundations, research design, and presentation of results in the shortest content. You should reorganize the description of the Abstract to bring it closer to the more critical content, rather than using word limits as a reason to refuse to add descriptions.

The abstract was revised as per the reviewer’s suggestion so that it included information relevant to background, methods, results and conclusions (although titles were not included as per journal format). The revised version sets up the rationale for the ADHD group, includes information on the analytic approach, defines acronyms, and highlights the preliminary nature of the results, as requested. We could not add more information to support the conclusions due to the word limit of 200. If there is specific information that the reviewer thinks could be removed and information that the reviewer wants added to the Abstract, we would be open to reviewing these edits.

Introduction

Overall: 1) The author directly deleted the sentence with doubt, which is not my purpose, but needs a more specific and convincing explanation; 2) I have no problem with the other reviewer appreciated the extensive literature review. Nevertheless, extensive literature reviews and lengthy extraneous descriptions are not in conflict, and I still expect a more concise and focused description.

The Introduction was substantially revised in the previous revision to address the reviewer’s feedback. The word count was reduced, information was re-organized, we reduced the focus on the adult literature and made clearer the rationale for inclusion of this literature, added general hypotheses, and made sure to more clearly build the rationale for the study and for the inclusion of the ADHD group. If the reviewer has specific feedback about changes they would like to see made, we would be happy to try to address these suggested changes.

Methods

Overall: (1) For explanations in the Author Response, please also describe them clearly in the manuscript; (2) Since the high loss of follow-up rate is a common problem of similar studies, the author should take it into account when calculating the sample size at the initial stage of the study and recruit a larger sample.

             Each of the reviewer’s comments were responded to and how they were addressed was noted. All changes that were made are in track changes. It is not possible to continue to recruit additional participants as the grant is finished, ethics for recruitment and study procedures are closed, etc. The power calculations did take into account the number of participants we needed to complete study procedures. We also made a clear statement in the limitations of the Discussion section about the sample size concerns (i.e., “It should also be highlighted that the current study had a modest sample size which was reduced further for some analyses. This likely impacted our ability to detect effects and limits confidence in our results. Therefore, the findings presented above should be thought of as hypothesis-generating as they provide preliminary evidence of altered homeostatic processes in children with ADHD and offer possible directions for future lines of inquiry.”)

Results

Overall: What makes the reader’s reading difficult is the confusing logic of expression and the lack of concise linguistic descriptions; it is not reasonable to fundamentally remove the research design, and follow-up still has its research relevance.

It is unclear how to address this concern. The manuscript has been read by a number of peers in this field, as well as all co-authors, and none indicated any problems with the logic. If there are specific examples, we would be happy to review these to see if we can make the information clearer.

Overall 

The author made some changes to the article, but it still did not address the main doubts. First of all, the author should give comprehensive consideration to the opinions of each reviewer. It is normal that different reviewers have different opinions. The author should not use another reviewer's opinion to reject my opinion, but should list more authoritative evidence to convince me. In addition, although this article is an exploratory study, the author did not explain in detail the problem of small sample size and high loss to follow-up, which did not convince me. Finally, the author is supposed to highlight the modified part to better represent the traces of modifications. 

             We are sorry that you did not find our responses to your feedback and the resulting changes to the manuscript to meet your expectations. We did take your feedback seriously and implemented the changes we thought addressed your concerns, when possible. We did extensively revise the Introduction based on your feedback. It is unclear why you feel we rejected your feedback given that we addressed each and every point to the best of our ability. All changes in the revised manuscript were made with track changes and were visible in the revision as track changes. Thank you again for your feedback which we believe resulted in a stronger revised manuscript, despite it not meeting your expectations.

We are submitting the second revised version as both a track change and clean version.  
